# Computational identification and experimental characterization of preferred downstream positions in human core promoters

**René Dreos**[1☯¤], **Anna Sloutskin**[2☯], **Nati Malachi**[2☯], **Diana Ideses**[2], **Philipp Bucher**[1,3]*, **Tamar Juven-Gershon**[2]*

**1** Swiss Institute of Bioinformatics (SIB), Lausanne, Switzerland, **2** The Mina and Everard Goodman Faculty of Life Sciences, Bar-Ilan University, Ramat-Gan, Israel, **3** School of Life Sciences, Swiss Federal Institute of Technology, Lausanne, Switzerland

☯ These authors contributed equally to this work.
¤ Current address: Center for Integrative Genomics, University of Lausanne, Lausanne, Switzerland.
* philipp.bucher@sib.swiss (PB); tamar.gershon@biu.ac.il (TJG)

**Data Availability Statement:** Source data are available from: promoter collections, ftp://epd.epfl. ch/ Public data used: ATAC-seq data for human

## Abstract

Metazoan core promoters, which direct the initiation of transcription by RNA polymerase II (Pol II), may contain short sequence motifs termed core promoter elements/motifs (*e.g.* the TATA box, initiator (Inr) and downstream core promoter element (DPE)), which recruit Pol II via the general transcription machinery. The DPE was discovered and extensively characterized in *Drosophila*, where it is strictly dependent on both the presence of an Inr and the precise spacing from it. Since the *Drosophila* DPE is recognized by the human transcription machinery, it is most likely that some human promoters contain a downstream element that is similar, though not necessarily identical, to the *Drosophila* DPE. However, only a couple of human promoters were shown to contain a functional DPE, and attempts to computationally detect human DPE-containing promoters have mostly been unsuccessful. Using a newly-designed motif discovery strategy based on Expectation-Maximization probabilistic partitioning algorithms, we discovered preferred downstream positions (PDP) in human promoters that resemble the *Drosophila* DPE. Available chromatin accessibility footprints revealed that *Drosophila* and human Inr+DPE promoter classes are not only highly structured, but also similar to each other, particularly in the proximal downstream region. Clustering of the corresponding sequence motifs using a neighbor-joining algorithm strongly suggests that canonical Inr+DPE promoters could be common to metazoan species. Using reporter assays we demonstrate the contribution of the identified downstream positions to the function of multiple human promoters. Furthermore, we show that alteration of the spacing between the Inr and PDP by two nucleotides results in reduced promoter activity, suggesting a spacing dependency of the newly discovered human PDP on the Inr. Taken together, our strategy identified novel functional downstream positions within human core promoters, supporting the existence of DPE-like motifs in human promoters.

lymphoblastoid cell line GM12878: Source data: GEO series GSE47753, samples GSM1155957, GSM1155958, GSM1155959, GSM1155960 Processed data: MGA series buenrostro13, sample GM12878|ATACseq|50K|short link to file: ftp://ccg. epfl.ch/mga/hg19/buenrostro13/GM12878_50K. oriented.sga ATAC-seq data for *Drosophila* wild type eye-antennal imaginal disc: Source data: GEO series GSE59078, sample GSM1426261 Processed data: MGA series dm6/davie15/, sample WT|FAIRE|Control link to file: ftp://ccg.epfl.ch/mga/ dm6/davie15/GSM1426261.sga Processed data, R scripts and results are available from https://github. com/talponer/humanDPE.

**Funding:** The work was supported by the Swiss Government (to PB) http://www.admin.ch and the Israel Science Foundation (no. 798/10 to TJG) https://isf.org.il. AS was supported by the Nehemia Levzion Scholarship and Bar-Ilan University President's Scholarship. Research funds available to the TJG from Bar-Ilan University https://www1. biu.ac.il/. The funders had no role in study design, data collection and analysis, decision to publish, or preparation of the manuscript.

**Competing interests:** The authors have declared that no competing interests exist.

## Author summary

Transcription of genes by the RNA polymerase II enzyme initiates at a genomic region termed the core promoter. The core promoter is a regulatory region that may contain diverse short DNA sequence motifs/elements that confer specific properties to it. Interestingly, core promoter motifs can be located both upstream and downstream of the transcription start site. Variable compositions of core promoter elements were identified. The initiator (Inr) motif and the downstream core promoter element (DPE) is a combination of elements that has been identified and extensively characterized in fruit flies. Although a few Inr+DPE -containing human promoters were identified, the presence of transcriptionally important downstream core promoter positions within human promoters has been a matter of controversy in the literature. Here, using a newly-designed motif discovery strategy, we discovered preferred downstream positions in human promoters that resemble fruit fly DPE. Clustering of the corresponding sequence motifs in eight additional species indicated that such promoters could be common to multicellular non-plant organisms. Importantly, functional characterization of the newly discovered preferred downstream positions supports the existence of Inr+DPE-containing promoters in human genes.

## Introduction

Regulation of eukaryotic gene expression is critical for diverse biological processes, including embryonic development, differentiation, cell cycle progression and apoptosis. Cellular signals that regulate gene expression affect many different factors and co-regulators, but the ultimate decision whether or not to initiate transcription occurs at the core promoter. The core promoter, which lies at the heart of transcription, is generally defined as the minimal region that directs the accurate initiation of transcription by RNA polymerase II (Pol II) [1–5].

There are three major modes of transcription initiation patterns: focused, dispersed and mixed [1–3,5–9]. Focused (also termed "sharp") promoters encompass from −40 to +40 relative to the transcription start site (TSS; referred to as +1), and contain a single predominant TSS or a few TSSs within a narrow region of several nucleotides. Focused transcription initiation is associated with spatiotemporally regulated genes. Because of the biological significance of regulated genes, focused initiation is the most studied mode of transcription initiation. Dispersed (also termed "broad") promoters contain multiple weak start sites that span over 50 to 100 nucleotides. Dispersed transcription initiation is associated with constitutive or housekeeping genes. Mixed (also termed "broad with peak") promoters combine the abovementioned modes by exhibiting a dispersed initiation pattern with a single strong transcription start site.

Interestingly, although the core promoter was previously regarded as a universal component of the transcription machinery, it is nowadays clear that core promoters differ both in their architecture and function [1,3,5,10–12]. In addition, the core promoter composition was demonstrated to affect transcriptional output, thus demonstrating the regulatory role of the promoter sequence itself [13–16].

Metazoan focused core promoters may contain short DNA sequences termed core promoter elements/motifs. These motifs, such as the TFIID-bound elements TATA box, initiator (Inr), downstream core promoter element (DPE), motif ten element (MTE) and the Bridge configuration, function as recognition sites for the basal transcription machinery that recruits Pol II and have a positional bias (reviewed in [1–5,17,18]). The function of the DPE, MTE and Bridge downstream motifs is exclusively dependent on a strictly-spaced functional Inr motif [19–22].

The DPE, MTE and Bridge motifs were discovered and extensively characterized in *Drosophila melanogaster* promoters [16,19–32]. Although the conservation of the DPE and MTE from *Drosophila* to humans was demonstrated, only a few human promoters were shown to be dependent on a functional DPE strictly located at positions +28 to +32, relative to the $A_{+1}$ of the Inr [20,33,34], and one review article even postulated that the DPE may be unique to *Drosophila* [3]. Nevertheless, as fruit flies are evolutionarily distant from humans, it is very likely that some human promoters contain a downstream core promoter element that is similar, but not identical to, *Drosophila* DPE.

TFIID is the first basal transcription factor that binds the core promoter and recruits Pol II and other basal transcription factors to initiate transcription [1,4,35–38]. The TAF1 and TAF2 subunits of TFIID were previously implicated in binding the downstream core promoter region [39]. Remarkably, the downstream region of the super core promoter (SCP), a synthetic promoter that includes the TATA box, Inr, MTE and DPE [14], exhibits a robust transcriptional output in multiple human cell lines [14,40], as compared to other commercially-available potent promoters. Mutating any of these 4 elements significantly reduces TFIID binding and the transcriptional output of the SCP [14,41]. This observation strongly suggests that the transcription machinery in human cells recognizes downstream positions conforming to the *Drosophila*-defined DPE and MTE motif sequences. Moreover, based on recent cryo-electron microscopy (cryo-EM), it was suggested that the SCP is bound by the TAF1, TAF2 and TAF7 subunits of human TFIID [42]. These findings imply that distinct human core promoters are recognized by the transcription machinery in human cells via specific nucleotides in the downstream core promoter region.

To identify preferred downstream positions in focused human core promoters, we designed a motif discovery strategy, using probabilistic partitioning algorithms, based on Expectation-Maximization model optimization.

This algorithm was applied to human and *Drosophila* core promoter regions comprising the base pairs from -10 to +40 relative to the TSS. Interestingly, we identified downstream overrepresented positions that resemble the *Drosophila* DPE motif. Available chromatin accessibility (ATAC-seq) footprints reveal that *Drosophila* and human Inr+DPE promoter classes resemble each other, especially in the proximal downstream region. In addition, human Inr+DPE promoters are more focused than other promoters' classes based on CAGE data. A reduced, but not completely absent, nucleosome positioning signal in human Inr+DPE promoters is detected - similar to the signal observed for TATA-box promoters. Clustering analysis of the identified sequence motifs in ten species using a neighbor-joining algorithm indicated that canonical Inr+DPE -containing promoters could be common to metazoan species. Using dual-luciferase reporter assays we demonstrate the contribution of the identified downstream positions to the function of several human promoters. Furthermore, we show that the spacing between the preferred downstream positions and the Inr motif is important for human core promoter activity, as demonstrated for *Drosophila* promoters. Taken together, our motif discovery strategy identified novel functional downstream positions in human core promoters, supporting the existence of DPE-like motifs in the downstream region of human promoters that may serve as recognition sites for human TFIID.

## Results

### Evidence for preferred downstream positions that resemble the DPE, in human promoters

The DPE motif is readily identified in *Drosophila* [16,19,20,22,24,26–30,32], and there is unquestionable evidence that *Drosophila* DPE motifs are recognized by the human transcription machinery *in vitro* and in multiple human cell lines [14,20,33,34,41]. Nevertheless,

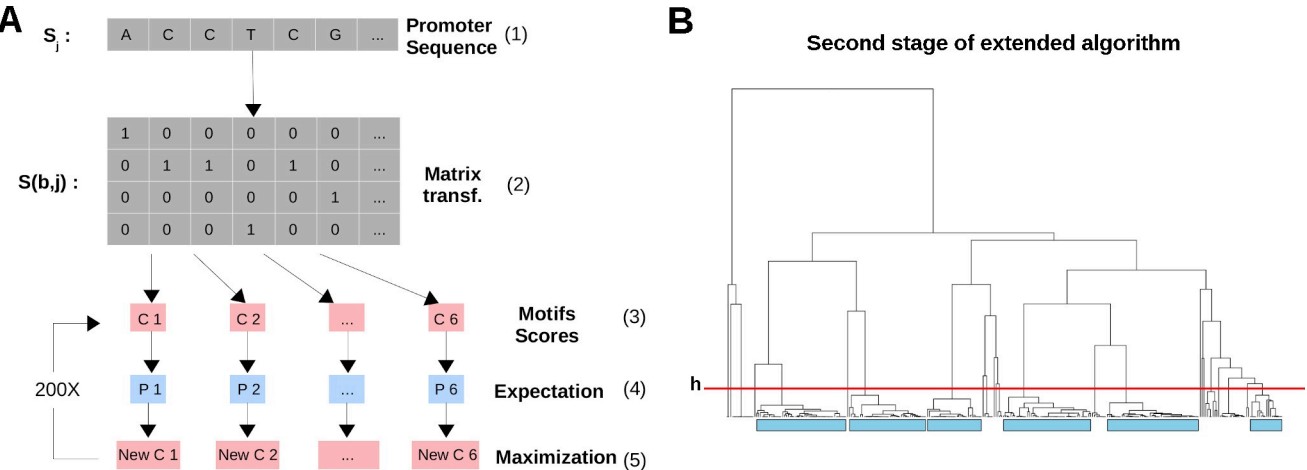

**Fig 1. EM algorithms implemented in this study.** (A) Diagram of the simple probabilistic partitioning basic algorithm. A promoter sequence $S_i$ (1) is transformed into a binary matrix $S(b,j)$ (2) following guidelines in [60] where each row represents one of the four bases $b$ (A, C, G and T). Each element of the matrix $S(b,j)$ has a value of 1 if the corresponding base is present at position $j$ in the sequence. The matrix is then scored against $K$ number of motifs (in this example K=6, C1 to C6) (3) to generate a probability score for each motif (P1 to P6) (4). In the first cycle, the motifs are generated using a random seeding strategy where the sequence probabilities follow a beta distribution. Next, each motif consensus is updated using the promoter sequences in conjunction with their probabilities (New C1 to New C6) (5). This cycle is repeated a number of times (in this example 200 times) to obtain the final motifs. (B) Probabilistic partitioning extended algorithm. All steps in A are repeated a number of times (in this example 50 times) to generate 300 motifs. These are then clustered hierarchically. The resulting tree is cut at a specific height ($h$, here at distance equal to 0.5) and the K nodes comprising the largest amount of motifs (identified by cyan rectangles) are retained and averaged to generate the final motifs.

attempts to computationally identify a corresponding sequence motif in human promoters have been controversial [3]. Applying the basic probabilistic partitioning algorithm illustrated in Fig 1A, we can easily identify a DPE motif in *Drosophila* promoters (Fig 2). Partitioning *Drosophila* promoters into three subclasses, we obtained one class containing both a canonical Inr and DPE motif (Class 1), a second one containing only an Inr motif (Class 2), and a third one containing a weak non-canonical Inr motif featuring G and A at about equal frequency at the TSS, which is preferentially flanked by T's on both sides (Class 3). Applying the same algorithm to human promoters, the results were somewhat different from the results of the run on *Drosophila* promoters: we identified a class containing a strong canonical Inr motif (Class 1) and another one containing a surprisingly similar weak non-canonical Inr motif (Class 2). A third identified class had almost no conserved base positions, except a weak preference for a purine at the TSS (Class 3). Strikingly, scanning human promoters using the DPE motif logo revealed a clear enrichment at the same downstream positions as in *Drosophila* promoters (S1 Fig). This result indicates that a downstream motif, which is similar to the *Drosophila* DPE, might be conserved to human promoters and potentially reflect its functional importance.

It is important to remember in this context that at least three human promoters, namely IRF1, CALM2 and TAF7 (TAFII55), were experimentally shown to have functional DPE motifs [20,33,34]. In line with this, human Class 1 promoters seemed to contain a very weak preference for nucleotides in positions +28, +29, which prompted us to develop a more refined algorithm. One potential limitation of the basic probabilistic partitioning algorithm is that it appears to have a tendency to split the input sequences into classes of similar sizes, as can be inferred from the frequencies presented in Fig 2. If we hypothesize that the DPE motif occurs only in a very small subclass of human promoters, the corresponding sequence motif may simply be hidden in one or several of the abundant subclasses shown in Fig 2. To test this hypothesis, we modified the basic algorithm to favor the discovery of low frequency classes with highly skewed base composition (Fig 1B, Methods section).

 

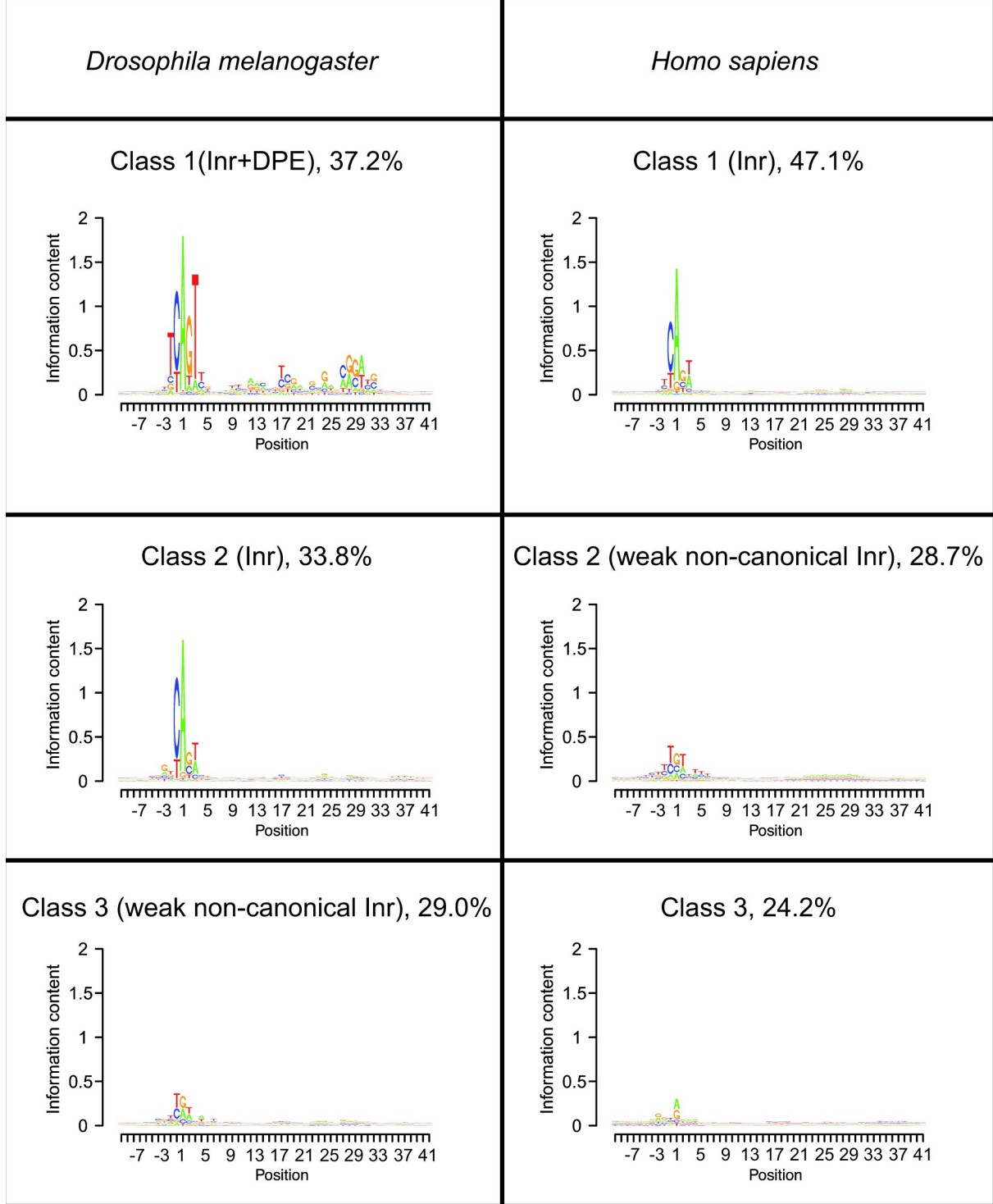

**Fig 2. Partitioning of promoter sequences using the basic probabilistic partitioning EM algorithm.** Three major classes with distinct core promoter compositions were identified within *Drosophila melanogaster* and human promoters. For each class, its frequency among the examined promoters is indicated. Numerical data are provided in S2 File.

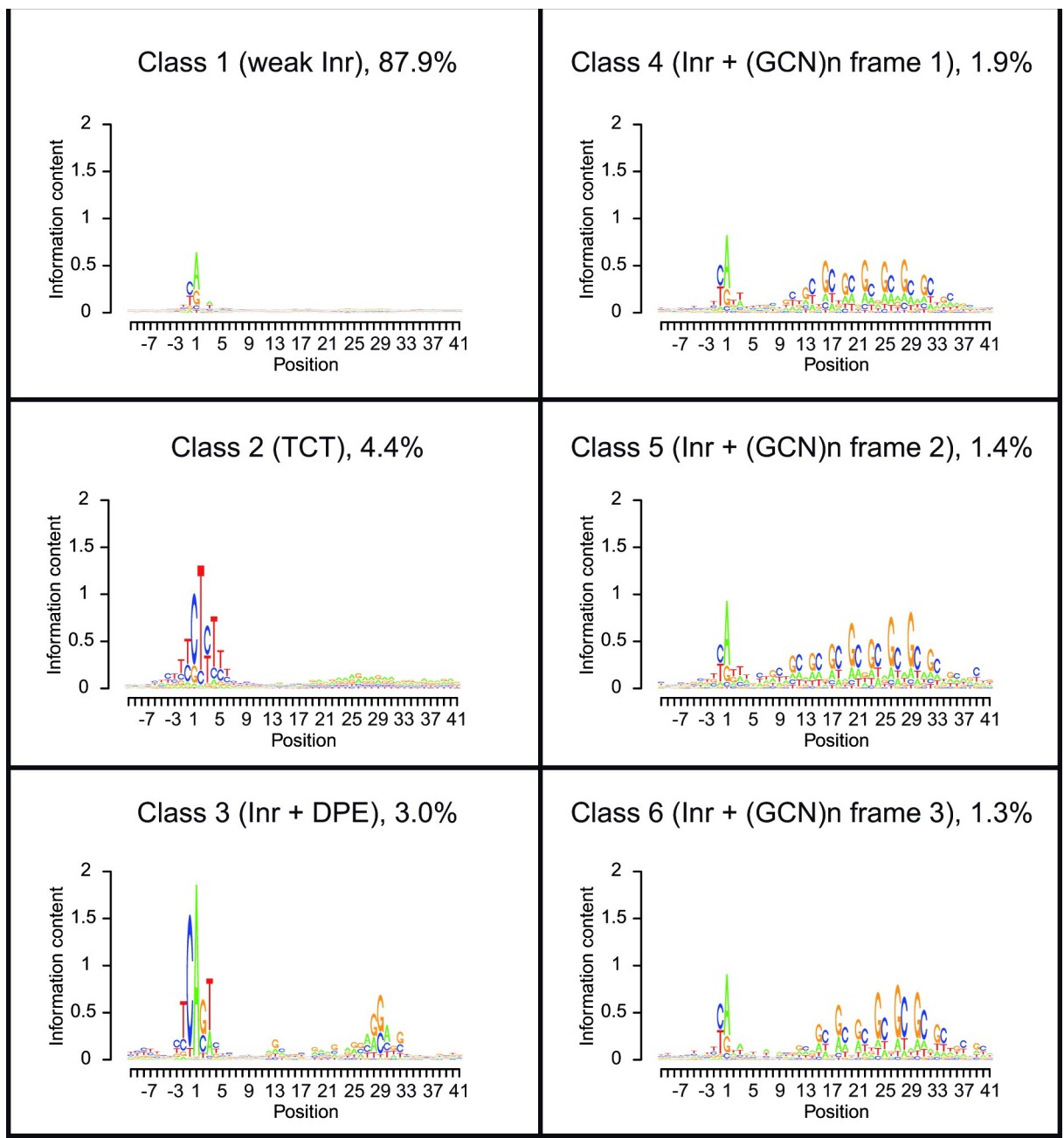

**Fig 3. Partitioning of human promoter sequences using the newly developed extended EM algorithm.** Six most frequent classes were identified within human promoters. The third most frequent class, which very much resembles the Inr+DPE class found in *Drosophila*, accounts for 3% of human core promoters. For each class, its frequency among the examined promoters is indicated. Numerical data are provided in S2 File.

Applying the new algorithm to human promoter sequences, a stable partitioning was achieved with 6 classes (Fig 3). The vast majority of promoters (87.9%) fall into a major class showing a very weak initiator motif, essentially consisting of a purine at the TSS preceded by a pyrimidine, previously termed a $YR_{+1}$ initiator [3,7,43]. This class is reminiscent of Class 1 obtained with the basic partitioning algorithm. The second most frequent class (4.4%) contains

another known element, the TCT motif [44], which is found in promoters of ribosomal protein genes and other genes related to translation. The third most frequent class (3.0%) very much resembles the Inr+DPE class found in *Drosophila*. In particular, positions 28-32 relative to the $A_{+1}$ of the Inr, show almost identical base preferences between the two species. The remaining three classes show the same trinucleotide-repeat pattern $(GCN)_n$ in three different frames relative to an initiator motif, consisting mostly of a purine at the TSS preceded by a pyrimidine. To our knowledge, this is a new pattern of unknown function. The 3 bp periodicity suggests a function in translation. Indeed, this hypothesis is further supported by the fact the trinucleotide repeats are preferentially preceded by in-frame translation initiation codons in all three classes (S1 Table).

In order to identify low frequency classes, which could have been missed with the basic algorithm, the new algorithm was also applied to *Drosophila* promoters, partitioning them into 6 classes (S2 Fig). Based on its abundance and motif pattern, we speculate that the majority class (88.9%) obtained by this run is a mixture of all three classes obtained with the basic algorithm (Fig 2). Class 4 (2.5%) shows an apparent alternative initiator motif at position +1, GGTCACACT, but virtually no sequence conservation elsewhere. To our surprise, this motif is virtually identical to Ohler Motif 1 [28], suggesting that a significant subset of Motif 1-containing promoters exhibits this motif exactly at the TSS and in the forward orientation. The other four classes are variants of the Inr+DPE class. In contrast to our expectations, no TCT and no trinucleotide repeat-containing classes were discovered. In summary, with regards to rare promoter classes, six-fold partitioning of human and *Drosophila* promoters highlights differences rather than commonalities between the two species.

To assess the robustness of the newly identified promoter classes, we performed bootstrapping. The complete promoter set was randomly resampled 10 times using the "sampling with replacement" method. The resampled promoter sets were then analyzed with the extended partitioning algorithm. To minimize the risk that a class is missed by chance, we retained the 10 rather than 6 most frequently found classes from each bootstrapping round. To quantify reproducibility, we recorded for each class in Fig 3 the Pearson correlation coefficient with the most similar subclass from each round (S3 Fig). The results are highly reassuring. Five of the six newly identified promoter classes (including Inr+DPE) are reproduced by all resampled data sets with a high correlation coefficient ($r > 0.8$). For class 6 (a GCN-repeat class), one (out of 10) of the bootstrapping rounds demonstrated low correlation with the newly identified class ($r = 0.26$).

In addition, we performed GO terms analysis on the *Drosophila melanogaster* classes identified by the basic probabilistic partitioning EM algorithm (Fig 2) and the *Homo sapiens* classes identified by the extended probabilistic partitioning EM algorithm (Fig 3) using PANTHER [45]. The canonical Inr and DPE motif-containing *Drosophila* Class 1 was found to be enriched for the regulation of biological processes and developmental processes (S4 Fig), as expected [24,32]. The GO term analysis of human Class 2, TCT enriched promoters, confirmed the enrichment of ribosomal and protein synthesis-related genes, as previously demonstrated [44]. The Inr+DPE motif-containing human Class 3 genes were found to be enriched for spindle assembly and organization, yet less significant. No GO terms enrichment was detected for either of the human classes 4 to 6, exhibiting the (GCN)n pattern in three different frames.

## The new computationally identified human Inr+DPE class closely resembles its *Drosophila* counterpart in terms of its DNA accessibility

The question whether a *Drosophila*-like DPE element exists in human (or in any other species) could also be debated from a biochemical perspective. In this case, one would have to show

that the human DPE discovered computationally in this study undergoes similar protein-DNA interactions as its well-characterized *Drosophila* counterpart. One way to approach this question is by looking at chromatin accessibility footprints [46]. Both DNase-seq and ATAC-seq assays provide genome-wide maps of DNA accessibility at single base resolution. The binding of a protein complex to a genomic region changes the exposure of individual phosphodiester bonds to DNase and transposase in the bound region. A characteristic footprint is revealed if sparse data from many DNA regions are super-imposed in an aggregation plot (as in [47]).

We evaluated ATAC-seq footprints for the most frequent promoter classes identified in *Drosophila* and human with regard to their capacity to discriminate between the computationally derived promoter classes (Fig 4A and 4B). Notably, compared to the other classes, the DPE-containing classes are highly structured in the +10 to +35 downstream regions. This suggests tight contacts with a specific protein surface, which do not occur in promoters lacking a DPE. Unsurprisingly, the ATAC-seq footprint of the human TCT class looks different from all other classes, especially at positions very close to the TSS.

The ATAC-seq footprints of the Inr+DPE promoter classes from the two species are not only highly structured but also similar to each other, in particular in the proximal downstream region (see detailed views in Fig 4C). In both species, local maxima appear at positions 1, 6, 10, 18, 20, 23, 28 and 33, while local minima appear at positions 7, 19, 21, 24, 29 and 34. Furthermore, a U-shaped valley is seen between positions 12 and 17.

To support these intuition-guided assessments in a more objective manner, we computed correlation coefficients of ATAC-seq footprints for all positions in the proximal downstream promoter regions for all pairs combinations of promoter classes (Fig 4D). Indeed, the two Inr +DPE classes show the highest correlation (*r*=0.89). Classes with a canonical or recognizable Inr (dm6_c1, dm6_c2, hg19_c1, hg19_c3) also show positive correlations among themselves, whereas the human TCT class (hg19_c2) negatively correlates with all but one class. In summary, our results confirm that the newly discovered human Inr+DPE class, identified by computational sequence analysis in a completely experiment-blind manner, closely resembles its *Drosophila* counterpart in terms of direct protein-DNA contacts.

## Association of human Inr+DPE promoters with other genomic features

In *Drosophila*, promoters with an Inr+DPE element constitute an abundant class associated with specific architectural and functional properties, including focused transcription initiation, a disordered nucleosome organization and over-representation in developmental genes [1,3]. Do their less frequent human counterparts show the same trends? Indeed, human Inr+DPE promoters are more focused than other promoters of other classes (Fig 5 and S2 Table). With regard to nucleosome architecture, nucleosome occupancy profiles derived from human MNase data (Fig 5, panels B,F) are not conclusive due to their noisiness (presumably related to the small number of promoters in this class). We tried to circumvent this limitation by quantifying the sequence-intrinsic nucleosome positioning signal as a proxy (Fig 5, panels C,G) for experimentally determined nucleosome architecture. To this end, we relied on a previously published method based on spectral analysis of dinucleotide periodicity [11]. Since TATA-box containing promoters were also reported to have disorganized nucleosomes, we included this class as a control. The results presented in Fig 5 show a reduced but not completely absent nucleosome positioning signal in human Inr+DPE promoters. The peak at period 10 is substantially lower than in the Inr-only classes from both species, but definitely not absent as in the *Drosophila* Inr+DPE promoter class. In summary, human Inr+DPE promoters only partially reproduce characteristic genomic features associated with their counterparts in *Drosophila*. While they are structurally similar, and (based on ATAC-seq footprints) most likely

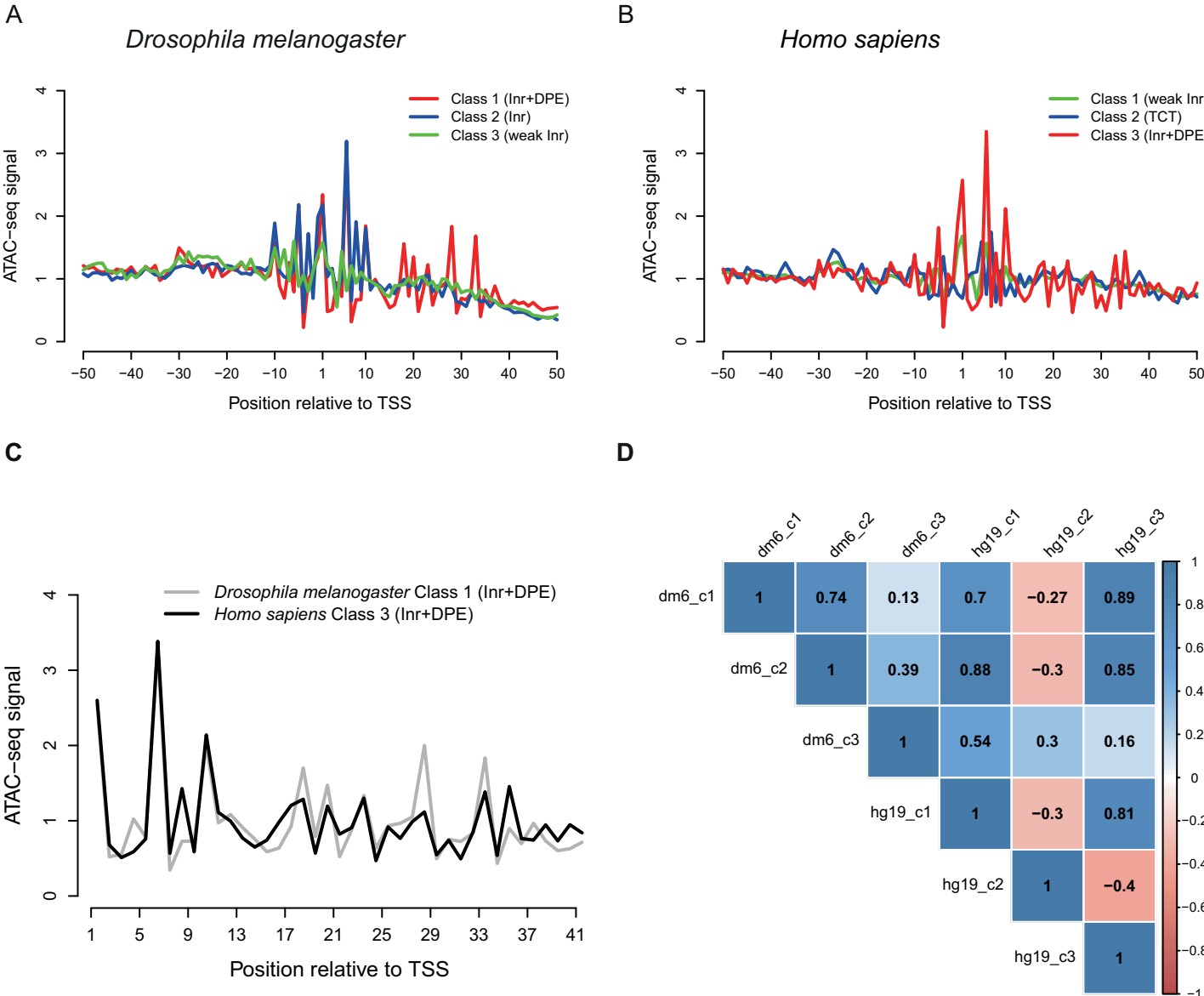

**Fig 4. ATAC-seq footprints of different promoter classes.** Single-base resolution ATAC-seq footprints are shown for the three most prominent promoter classes (Figs 2 and 3) in *Drosophila* (A) and *Homo sapiens* (B). (C) Enlarged footprints of the proximal promoter downstream regions of the two Inr+DPE classes are combined in one plot to facilitate cross-species comparison. (D) Pearson correlation coefficients computed from the same proximal promoter regions are shown for all possible pairs of promoter classes. The ATAC-seq signal displayed on the vertical axis of all line plots is expressed as fold enrichment over the local mean (*i.e.* for each line, the points were first divided by their mean before plotting). Numerical data are provided in S2 File.

engage in direct contact with homologous proteins, they may have functionally diversified in the vertebrate and insect lineages.

We were also wondering whether human DPE promoters are preferentially co-occurring or avoiding other sequence motifs. With regard to the TATA-box, which is associated with similar architectural features as the DPE motif, we observe some degree of avoidance (S2 Table), indicative perhaps of functional redundancy. Nevertheless, co-occurrence of the two elements remains quite common, suggesting that the motifs can still co-operate or substitute for each other in the same promoter. An earlier paper [48] reported strong over-representation of the

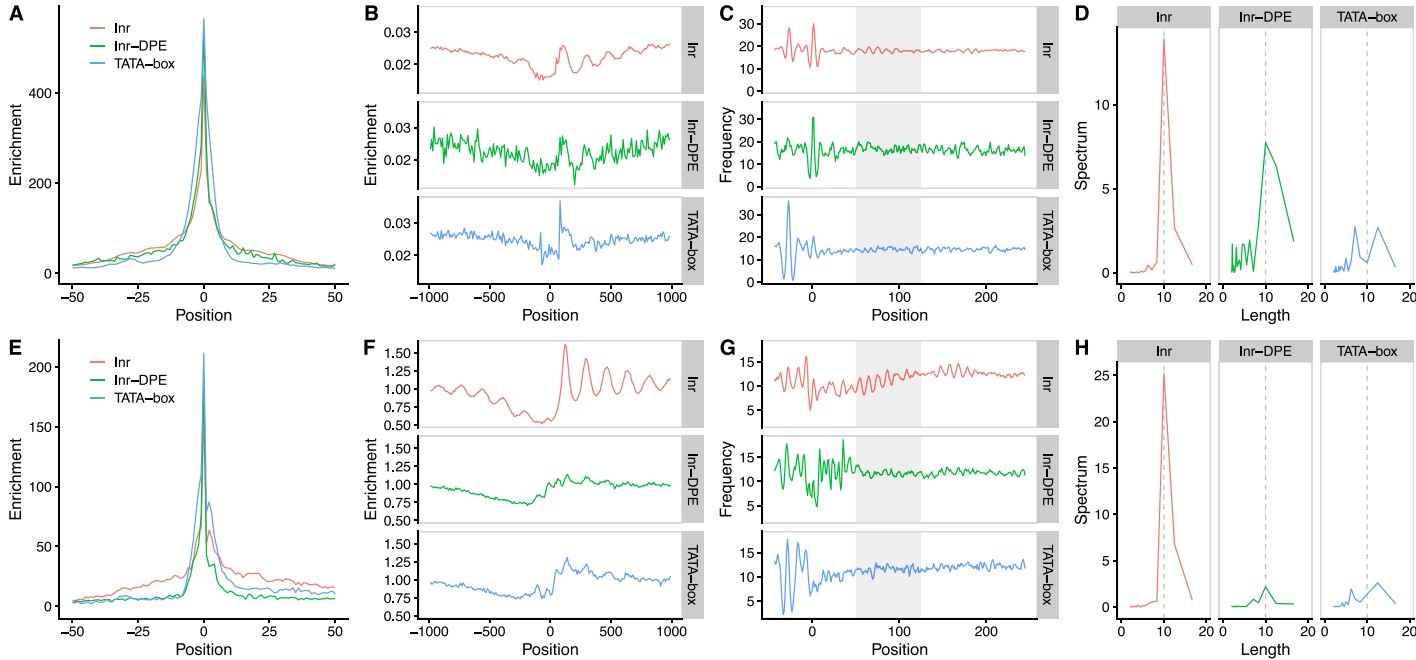

**Fig 5. Promoter classes architecture.** For *Homo Sapiens* (A-D), Inr and Inr-DPE classes correspond to Class 1 and Class 3 as in Fig 3, respectively. For *Drosophila melanogaster* (E-H), Inr class correspond to Fig 2 Class 2 and 3, while Inr-DPE class corresponds to Class 1 of the same figure. TATA-box annotation was derived from the EPD database. CAGE distribution around *Homo sapiens* (A) and *Drosophila melanogaster* (E) transcription start sites (TSS) stratified by presence of core promoter elements. CAGE data was retrieved from the FANTOM consortium [66] and Machibase [67] for *Homo sapiens* and *Drosophila melanogaster*, respectively. To reduce the impact of highly expressed promoters, CAGE tags mapping to the same location have been capped to 1000. The presence of the DPE element has a stronger effect on *D. melanogaster* promoters in focusing the initiation, compared to human. (B and F) Nucleosome distribution around *Homo sapiens* (B) and *Drosophila melanogaster* (F) TSSs stratified by the presence of promoter elements as in A and E. Inr promoters show an open chromatin configuration in both organisms, whereas TATA-box and Inr-DPE promoters demonstrate a closer conformation. (C and G) Nucleosome favoring signals as described in [11] around *Homo sapiens* (C) and *Drosophila melanogaster* (G) promoters, stratified as in A and E. For both organisms, it is evident that a strong 10bp periodic signal present in Inr promoters is strongly reduced by the presence of the Inr-DPE or the TATA-box. (D and H) Fourier transform to quantify the periodic signal strength is shown in C and G (area used in this analysis is marked by gray boxes). Numerical data are provided in S2 File.

YY1 motif in the proximal downstream region of human promoters, approximately at the same location where the DPE element occurs. We thus analyzed the positional distribution of the YY1 motif in the human promoter classes identified in this work (S5 Fig). Here, the TATA-box containing (TATA+) and Inr+DPE promoters show contrasting trends. While the YY1 motif is virtually absent in TATA+ promoters, it is over-represented in all specialized minor promoter classes (TCT, Inr+DPE, poly-GCN 1,2,3). The motif occurrence profiles show 2-3 peaks separated by about 10 bp, indicative of one-sided binding to the double helix. Interestingly, the Inr+DPE class shows the clearest 10-bp periodicity, suggesting that promoters of this class are tightly bound by a large protein complex on the other side of the DNA. This scenario would be entirely compatible with the prevalent view that Inr-DPE promoters make direct DNA-protein contacts with components of the TFIID complex. In summary, our motif co-occurrence analysis revealed additional distinctive features of human Inr+DPE promoters, further justifying their status as a separate promoter class.

It is possible that nucleosome positioning, among other molecular mechanisms, contributes to the transcriptional output of DPE-containing promoters. Nevertheless, our observations indicate that the DPE sequence motif affects transcriptional activity of DPE-dependent promoters.

## DPE-like motifs in other species

The finding that the Inr+DPE element was present in human promoters, raised the intriguing hypothesis that it could be more widespread and might occur in other species, perhaps even beyond the metazoan kingdom. To this end, we applied the EM partitioning algorithm with K=6 to promoters from eight additional species, including plants (*A. thaliana* and *Z. mays*) and fungi (*S. cerevisiae* and *S. pombe*). To visualize the relationship between all promoter classes obtained in this way (including those from *Homo sapiens* and *Drosophila*), we clustered the corresponding sequence motifs using neighbor-joining. The resulting tree (Fig 6) was composed of three distinct domains: two clades (colored blue and red) and a middle ground (green) comprising multiple branches originating from nodes close to the tree center. To relate these domains to the motifs shown in Figs 2 and 3, we included consensus logos for each domain, which were obtained by averaging over the base probabilities of all motifs from each domain. Clearly, the sequence logo of the red sub-tree resembles the *Drosophila* Inr+DPE promoter class shown in Fig 2. The green and blue domains corresponded to CA and TG variants of the basic YR initiator motif, respectively. We noted that *D. melanogaster* Inr+DPE motifs ((Dm).m1, m3, m4, m5, m6) have close neighbors from all metazoan species (*H. sapiens* (Hs). m3; *M. musculus* (Mm).m2; *D. rerio* (Dr).m2, m3, m5, and m6; *A. mellifera* (Am).m2, m3, m4, m5 and m6; *C. elegans* (Ce).m4 and m6), while none of them are from species outside the metazoan kingdom, like plant and yeast. Taken together, the aforementioned observations strongly suggest that canonical Inr+DPE promoters could in fact be common to all metazoan species, and absent outside the metazoan kingdom.

## The identified downstream positions are functional in HEK293 cells

In parallel to our computational efforts to confirm the existence of a human DPE element and to characterize its sequence determinants, we carried out experiments to test whether previously identified critical downstream positions are indeed functional. An initial list of 20 potentially functional DPE-containing human core promoters was compiled based on promoter shape, average expression and the absence of a TATA-box. To narrow down the selection towards experimental validation, we applied the ElemeNT algorithm [49] to detect possible initiator and DPE motifs, based on PWMs constructed using experimental work in *Drosophila* [49]. We also verified that the promoters lack a TATA-box upstream of the examined region, and ensured that the initiation type is sharp (S6 Fig). Finally, two candidate core promoters were chosen for experimental analysis, namely LRCH4 (Leucine Rich Repeats And Calponin Homology Domain Containing 4) and ANP32E (Acidic Nuclear Phosphoprotein 32 Family Member E).

Notably, the prominent positions in the newly identified human downstream motif (Fig 3, Class 3) are G nucleotides at positions +28 and +29 (relative to the $A_{+1}$ position of the relevant initiator motif). Moreover, a sequence bias at +24(G) (relative to the $A_{+1}$ of the Inr) was previously observed and experimentally shown to contribute to the function of *Drosophila* DPE-containing promoters [26]. Thus, we focused on 3 preferred downstream positions (+24, +28 and +29 relative to the $A_{+1}$ position of the relevant initiator motif), mutating each of them from G to T nucleotide (mPDP version; exact sequences provided in Table 1). These substitutions were based on prior knowledge regarding functional downstream positions in *Drosophila melanogaster* promoters [30,49].

We generated both WT and mPDP constructs (Table 1), and tested them using dual-luciferase assays in HEK293 cells (Fig 7A). Strikingly, the substitution of the 3 positions was sufficient to reduce LRCH4 and ANP32E reporter levels to either 0.6 or 0.75-fold relative to the WT promoter, respectively.

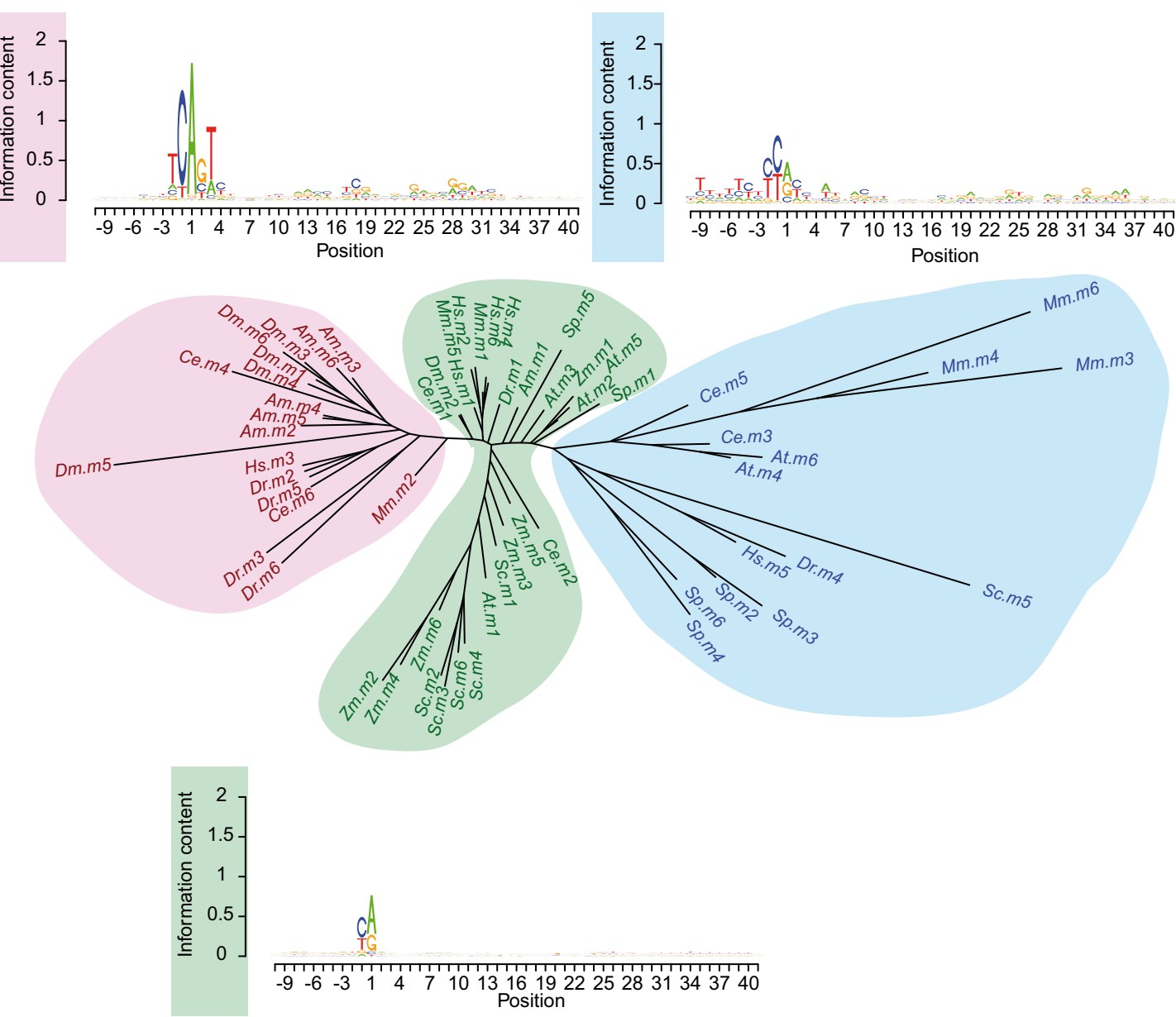

**Fig 6. Neighbor joining tree of motifs found in the promoter region of 10 species.** Global NJ tree obtained by clustering 6 motifs (identified using the presented EM algorithm, see Methods for detail) in 10 species (*H. sapiens*; *M. musculus*; *D. rerio*; *C. elegans*; *D. melanogaster*; *A. mellifera*; *A. thaliana*; *Z. mays*; *S. cerevisiae*; *S. pombe*). The tree is composed of two main clades (highlighted in red and blue) and a middle ground (green) containing several small branches originating from nodes close to the center. The consensus sequence of each clade is plotted alongside it. The Inr+DPE cluster (red) does not contain plants nor fungi motifs, highlighting the idea that the Inr +DPE element combination is present only in metazoa. The green and blue branches are variations of the basic YR motif.

We next sought to examine additional candidates, to gain a better understanding of the preferred downstream positions. Since the reduction in the LRCH4 reporter activity was more pronounced than that of the ANP32E gene (p-value 0.016) (as may have been expected based on the ElemeNT score, Table 1), we used the characteristics of LRCH4 as a reference. To this end, we started from a broader list of potentially-functional promoters. The resulting list was analyzed using ElemeNT, with the DPE score required to be >0.2 and accompanied by a Bridge element, similarly to LRCH4. The absence of a TATA-box was verified as well. We next

**Table 1. Sequences used for testing activity of identified downstream positions.**

| Name | Cloned promoter sequence | Class 3 score (EM) | | DPE score (ElemeNT) |
|---|---|---|---|---|
| | | log-odds | Z-score | |
| LRCH4 | cggtcccg*tcagtca*ggcagcgggagccgccgg**G**agc**GG**atggcggcggc | 11.24 | 4.96 | 0.2494 |
| ANP32E | atggaggc*tcagtct*ctgagcagccattgaagg**G**gaa**GG**aactgcgggtg | 13.58 | 5.26 | 0.0278 |
| CKS2 | tgcggtcg*ttagtct*ccggcgagttgttgcctg**G**gct**GG**acgtggttttgt | 7.22 | 4.09 | 0.8182 |
| CELF1 | ggggtgtt*ctgctct*ggcggcagcggcagcggc**G**gcg**GG**acgcggaggctc | -0.20 | 2.56 | 0.2425 |
| CTSA | catgactt*ccagtccc*cgggcgcctcctggaga**G**caa**GG**acgcgggggagc | 8.27 | 4.04 | 0.2425 |

Mutated positions are marked in bold and UPPERCASE (G>T substitutions). Initiator and DPE elements, as detected by the ElemeNT algorithm, are italicized or underlined, respectively. Since log-odds scores computed with the human Class 3 motif (Fig 3) are strongly influenced by base composition, we additionally computed Z-scores by shuffling each promoter sequence 100 times (see Methods).

applied a new additional cut-off expression levels criterion (based on HEK293 CAGE data generated by FANTOM5 consortium) - higher than or similar to those of LRCH4 - to select candidate promoters that would likely be expressed in our experimental system. Moreover, transcription initiation pattern (sharp or broad) was manually determined using the EPDnew website for each examined gene, based on the distribution of CAGE tags around the reported transcription start site.

Using the above guidelines, we chose 3 additional unrelated promoters to be tested, namely, CKS2 (CDC28 Protein Kinase Regulatory Subunit 2), CELF1 (CUGBP Elav-Like Family Member 1) and CTSA (Cathepsin A). Using dual-luciferase reporter assays in HEK293 cells, we discovered that reporter activities of the mutated CKS2 and CELF1 were reduced to either 0.6 or 0.8-fold relative to the WT promoter, respectively (Fig 7B). However, the luciferase reporter activity of the mPDP version of CTSA was not significantly lower than the WT version. Notably, this may result from the transcription initiation pattern of CTSA, which was slightly less focused than LRCH4, ANP32E, CKS2 and CELF1 (S6 Fig). Taken together, using the

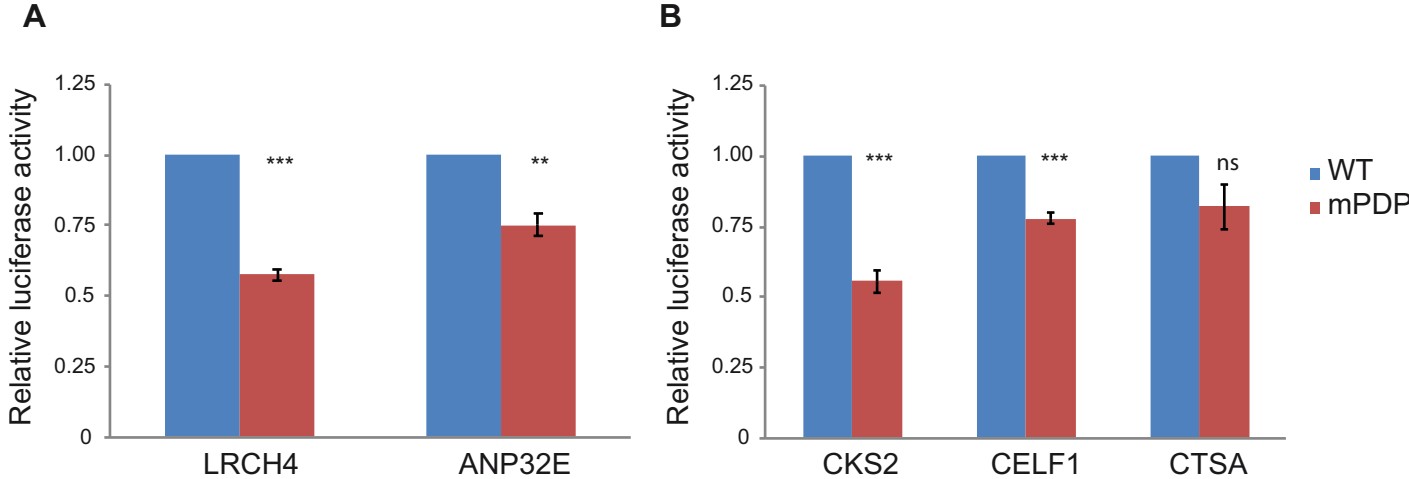

**Fig 7. The preferred downstream core promoter positions (PDP) are functional in HEK293 cells.** Results indicate the fold change in the WT versus mPDP version of the relevant promoter, tested by dual-luciferase assays in HEK293 cells. Each experiment was performed in triplicates, results represent 4-6 independent experiments ±SEM. ***p<0.001, **p<0.01, ns- not significant, calculated using Student's t-test. Two candidate genes, LRCH4 and ANP32E (A), were first chosen based on their core promoter composition and conservation, as discussed in the Results section. (B) As the reduction in the LRCH4 reporter activity was more pronounced than that of the ANP32E gene, the characteristics of LRCH4 promoter were used as a reference (see Results section for the exact criteria), and the promoters of CKS2, CELF1 and CTSA were chosen for experimental examination. Numerical data are provided in S2 File.

described EM algorithm and reporter assays in HEK293 cells, we identified a preference for conserved downstream positions within natural human core promoters with sharp transcription initiation patterns, and demonstrated that they are functional.

## The identified downstream positions are dependent on the spacing from the Inr

In order to test whether the identified downstream positions are canonical core promoter elements that, similarly to the *Drosophila* DPE, are dependent on the spacing from the Inr, we generated multiple mutants of the of LRCH4, ANP32E, CKS2 and CELF1 promoters, in which two nucleotides were either deleted or added (m2 or p2, respectively) in positions 10 or 18 relative to the $A_{+1}$ position of the TSSs. Using dual-luciferase reporter assays in HEK293 cells, we detected significantly reduced activities of LRCH4 promoters in which 2 nucleotides were either deleted or added at the tested positions (Fig 8). Although deletion of 2 nucleotides in position 10 of the ANP32E, CKS2 and CELF1 promoters did not result in reduced activity, significantly reduced activities were detected for mutants in which 2 nucleotides were either deleted or added at position 18, and when 2 nucleotides were added in position 10. Interestingly, position +10 of DPE-containing promoters is located in a DNase I protected region [14,26], suggesting that this stretch contains a yet-uncharacterized sequence motif that could be independent of the Inr, and might contribute to the transcriptional activities. Therefore, the differences in the transcriptional output for position 10 mutants were expected to be less profound than for position 18 mutants. Indeed, deletion of 2 nucleotides at position +10 affected the reporter activities to a lesser extent compared to position +18. By and large, the effects of these addition/deletion mutations argue in favor of a spacing dependency of the newly discovered PDP on the Inr, and against the possibility that these PDP merely serve as a binding site for a sequence-specific transcription factor that is not typically associated with core promoters.

We also examined whether two consecutive G nucleotides outside the PDP could result in reduced activities, similar to the observed mPDP activities. To this end, we mutated 2 consecutive G nucleotides to T nucleotides (mGG) in the vicinity of the PDP in the LRCH4 (at +35-36), ANP32E (at +38-39), CKS2 (at +34-35) and CELF1 (at +38-39) promoters. Interestingly, only the mGG version of the CKS2 promoter did not display reduced activity. Thus, the specific context of core promoter elements may have variable effects, as previously demonstrated (see [50], for example). The effect of the downstream GG dinucleotide might represent another, yet uncharacterized, contribution of the downstream core promoter to transcriptional regulation.

## Discussion

The presence of downstream core promoter positions within human promoters that are transcriptionally important has been a matter of controversy in the literature. Although the DPE was originally reported as conserved from *Drosophila melanogaster* to humans [20], and additional studies identified functional downstream core promoter motifs in human promoters [33,51], one publication suggests that the DPE motif is *Drosophila melanogaster*-specific [3], whereas another bioinformatics analysis indicated that ~25% of human promoters contain a sequence that matches the consensus of *Drosophila* DPE [52]. It should be noted, however, that the latter study did not account for the strict spacing dependency between the DPE and the Inr.

Nonetheless, ample evidence exists showing that the downstream region is an important regulator of transcriptional output in humans. The super core promoter (SCP), containing the TATA-box, initiator, MTE and DPE core promoter motifs, exhibits a robust transcriptional

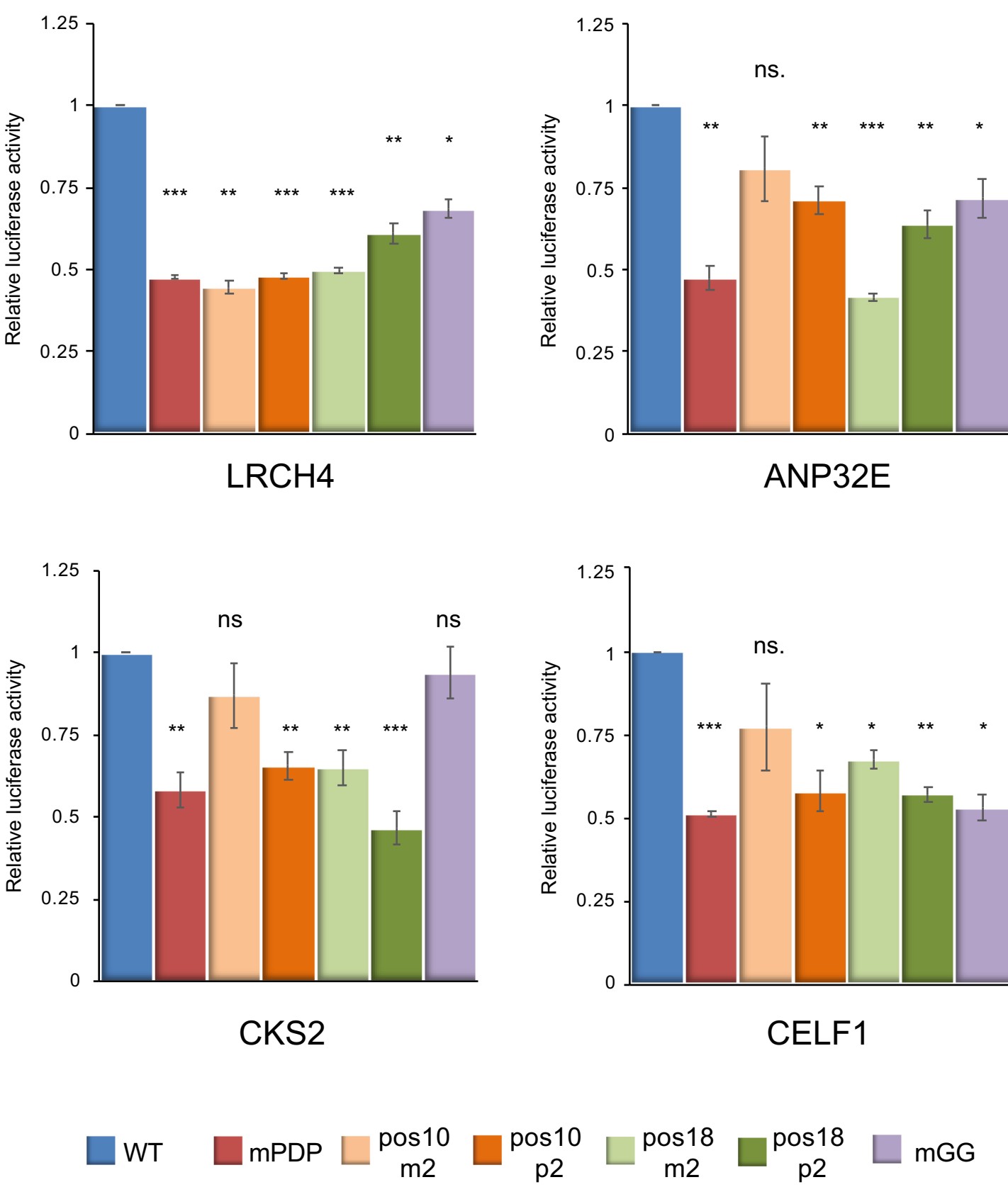

**Fig 8. The activities of the LRCH4, ANP32E, CKS2 and CELF1 promoters in HEK293 cells are dependent on the spacing between the Inr and the PDP.** Results indicate the fold change in the WT versus the mutant versions (mPDP, deletion or addition (m2 or p2, respectively) of 2 nucleotides in positions 10 or 18 relative to the $A_{+1}$ position of the TSSs, or mutation of 2 consecutive G nucleotides in the vicinity of the PDP to T) of the indicated promoters, tested by dual-luciferase assays in HEK293 cells. Each experiment was performed in triplicates, results represent 3-5 independent experiments ±SEM. ***p<0.001, **p<0.01, ns- not significant, calculated using Student's t-test. Numerical data are provided in S2 File.

output in human cells, as compared to other commercially-available potent promoters [14,40]. Mutating any of these elements significantly reduced the transcriptional output of the promoter [14], suggesting that the transcription machinery in human cells recognizes the DPE. Moreover, human TFIID is associated with the downstream core promoter area of the SCP [41,42,53], and both TFIID subunits TAF1 [42,53] and TAF2 [42,53,54] bind the downstream core promoter region.

The aim of our study was to search for a DPE-like core promoter motif in human promoters. In line with previous studies [3,7,43], our analysis showed that the majority of human promoters contain a $YR_{+1}$ initiator (Fig 3, Class 1). Importantly, using the extended EM algorithm, we discovered a novel class of human promoters containing an Inr and a downstream sequence motif that resembles the *Drosophila* DPE (Fig 3, Class 3). Unlike *Drosophila* DPE-containing promoters that account for more than a third of promoters (Fig 2, Class 1), human Class 3 promoters account for 3% and were not enriched for developmental processes or for biological regulation.

Interestingly, we did not identify an enrichment of human Inr and MTE (motif 10 element)-containing promoters. The MTE motif was first inferred from computational analysis of *Drosophila* promoter sequences [28]. The motif was originally defined by an algorithm allowing for extensive distance variation relative to the TSS. Its functional significance was later demonstrated in both *Drosophila* and human gene expression systems, using promoters from both species [21]. In that study, the MTE is presented as a core promoter element with a consensus sequence CSARCSSAACGS that occurs between positions +18 to +29, overlapping with the DPE motif by two base pairs. Similar to the DPE motif, it was reported that the MTE function is strictly dependent upon a functional Inr, and is involved in interaction with TFIID [21,22]. Furthermore, although it was defined as a distinct element, a synergy between the MTE and the DPE was demonstrated. The *Drosophila* Class 1 sequence logo that was detected using our algorithm supports C at position +18, R at +22, and CGS at +27-29. We further note an additional conserved Y at position +17, just preceding the reported MTE region.

Further examination of the downstream region revealed additional TFIID-interacting sub-regions, comprised of +18-22 and +30-33, termed Bridge [22]. The Bridge element was demonstrated to support, but not fully-restore, DPE-dependent transcription [30]. It was recently proposed that the downstream core promoter region might be a single functional unit (resembling the "Ohler-defined DPE", [28]) [55]. We compared our sequence motifs for the *Drosophila* and human Inr+DPE promoter classes to the "functional" MTE motifs of two *Drosophila* promoters (*Tollo* and CG10479) derived by single-base mutational analysis [22]. To make the motifs visually comparable, we converted mutational analysis data for each promoter into a corresponding sequence logo by dividing the relative transcriptional activities of each base at a given position by the sum of the transcriptional activities at the same position. We further computed Pearson correlation coefficients for all logo pairs, in order to assess similarity in a more objective manner (S7 Fig). By visual inspection we note a good agreement between the functional MTE motif of the CG10479 promoter and our computationally derived motif for the *Drosophila* Inr+DPE promoter class. This intuitive judgment is supported by a high Pearson correlation coefficient of 0.83. The functional MTE motif for *Tollo* shows more divergence with regard to both the CG10479 functional motif and the computationally derived Inr+DPE

motif. Not surprisingly, both functional motifs show better correlation with the *Drosophila* than with the human Inr+DPE motif. We further note a high correlation coefficient of 0.74 for the two computationally derived motifs, suggesting that the two species share conserved sequence determinants not only within the canonical Inr and DPE motifs, but also in the region between them.

A critical reader could question our results by arguing that we modified our extended partitioning algorithm to obtain the desired result. A general problem with partitioning and other unsupervised machine learning approaches is that the result cannot be assessed in terms of accuracy. We thus can only argue that the classification we obtain with the extended partitioning algorithm is biologically plausible or meaningful. The nature of the other four simultaneously discovered low-frequency promoter classes gives us assurance in this respect. Class 2 perfectly matches a previously reported promoter class, characterized by the presence of a TCT motif and its association with genes involved in translation. The other three minority classes strikingly resemble each other in that they contain the same trinucleotide repeats in three different frames relative to the TSS. These highly unusual properties make it unlikely that these classes are collateral noise of an algorithm specifically designed and fine-tuned to discover another promoter class.

The weak TGT motifs observed with the basic algorithm ([Fig 2](), *Drosophila* Class 3 and human Class 2), which are reminiscent of the previously described TGT motif [56], were not detected using the extended EM algorithm ([Fig 3]()). Notably, weak motifs in general, may reflect the presence of additional or a mixture of sub-classes of promoters.

Importantly, we demonstrate the contribution of the 3 G nucleotides, located at positions +24, +28 and +29 relative to the $A_{+1}$ position, to the function of four natural human promoters. Using luciferase reporters driven by minimal promoter constructs (-10 to +40) in HEK293 cells, we demonstrated that changing G nucleotides at these positions to T significantly reduces the transcriptional output to 0.5-0.8 fold, as compared to the WT promoters. This is a substantial effect on enzymatic reporter activities, considering the fact that only 3 nucleotides in a non-Inr region of the minimal promoters were substituted. Remarkably, the reduced reporter activities of promoters in which the spacing between the Inr and the DPE was altered by addition or deletion of 2 nucleotides, largely suggest that, similarly to the *Drosophila* DPE, the newly discovered PDP depends on spacing from the Inr. It also disfavors the possibility that the PDP serves as a binding site for a sequence-specific transcription factor that is not normally associated with core promoters.

Our analysis shows that the *Drosophila* and human DPE are similar, but not identical in all respects. While the *Drosophila* DPE is a robust driver of gene expression, the identified preferred positions among human promoters, although important, are probably used for "fine-tuning" of transcriptional activity. Interestingly, STARR-seq analysis demonstrated that *Drosophila* Inr+DPE promoters are associated with a distinct class of enhancers [15]. The association of human Inr+DPE promoters with a specific subset of enhancers remains to be determined.

During the preparation of the manuscript, we became aware of a comprehensive work from the Kadonaga lab [57], which used machine learning to generate predictive models to analyze human Pol II core promoters and identified a downstream promoter region (DPR) spanning from +17 to +35, which contributes to the transcriptional output of a fraction of human promoters. Reassuringly, the positions identified in our study highly match specific positions within the DPR identified by the Kadonaga lab, which supports the concept of a single functional downstream unit [28,55]. Moreover, different approaches to identify the important downstream positions were taken; while we started from bioinformatics analysis and then tested naturally-occurring minimal promoters, the Kadonaga lab has first used massively

parallel reporter assays (MPRA) of an extensive library composed of randomized version of the downstream region, using a specific promoter backbone. Moreover, the experiments were performed in two different cell lines, using different readout as the outcome, either the indirect luciferase reporter activity (this study) or the RNA output itself, using either RNA-seq or primer extension analysis [57]. Surprisingly, the two independent approaches identified functional downstream positions/region within the ANP32E promoter. Moreover, we ran the support vector regression (SVRb) model that was generated using *in vitro* transcription [57] on the +17 to +35 sequences of the wt and mutant promoters identified using the EM algorithm (S3 Table). Overall, our computational model was successful in making similar predictions (correlation coefficient ~0.78) as the SVRb model that used experimentally-based training data. Thus, both independently-performed studies complement each other, strengthening the notion that the downstream core promoter region contributes to transcriptional regulation of human promoters. Our mutational analysis highlights the importance of three specific nucleotides for the transcriptional output, as well the spacing requirement between the preferred downstream positions and the Inr motif, reminiscent of the *Drosophila* DPE.

To conclude, specific positions within the downstream core promoter region of human promoters are important for the transcriptional outcome; thus transcriptional regulation of human promoters via the downstream region is an important regulatory mechanism, likely conserved among metazoans but absent in other eukaryotes.

## Methods

### Promoter sets

The promoter sets and the corresponding dominant TSS positions were taken from EPDnew [58]: version 5 for *H. sapiens* and *D. melanogaster*; version 2 for *M. musculus*, *A. thaliana* and *S. cerevisiae*; version 1 for all other organisms studied. EPDnew promoter collections have been validated by hundreds of high-throughput sequencing experiments (*i.e.* CAGE), giving a very high confidence in identifying the correct transcription start site. For each gene, the promoter that was validated by the largest number of experiments was selected as the representative. This gave very high confidence for the positions of the initiation sites, and reduced the probability of selecting promoters used only in particular cell lines and/or conditions. Moreover, to reduce possible sequence bias by coding sequences, promoters that had translation start sites within the first 40 bases were discarded.

### Probabilistic partitioning basic algorithm

In its basic structure, the algorithm is identical to the Expectation-Maximization (EM) algorithm presented in [59], which was originally designed for partitioning sets of genomic regions based on ChIP-seq data and represented as count data (integer) vectors and is described in Fig 1A. The adaptation to sequence data requires some modifications described below.

In the following, we adhere to the notation used in Stormo's review on specificity models of protein-DNA interactions [60]. Sequences of length $N$ denoted $S_i$ are represented as binary matrices with four rows corresponding to the bases A, C, G and T, and $N$ columns corresponding to successive positions in the sequence. A matrix element $S_i(b,j)$ has a value of 1, if base $b$ occurs at the $j$th position of sequence $i$, and a value of zero otherwise. A class $C_k$ is represented by a matrix of the same dimensions as the sequences, plus its occurrence probability $p_k$. A matrix element $C_k(b,j)$ contains the probability that base $b$ occurs at the $j$th position of a

sequence belonging to class $k$. The probability of sequence $S_i$ given class $C_k$ is then given by:

$$P(S_i|C_k) = \prod_{b,j} C_k(b,j)^{S_i(b,j)} \tag{1}$$

The formula for computing the probability of class $C_k$ given sequence $S_i$ remains unchanged:

$$P(C_k|S_i) = \frac{p_k \cdot P(S_i|C_k)}{\sum_{k'} p_{k'} \cdot P(S_i|C_{k'})} \tag{2}$$

Using these probabilities, the base probability matrix for class $C_k$ is updated in 2 steps:

$$C^*_k(b,j) = \frac{\sum_i P(C_k|S_i)S_i(b,j)}{q_b} Z_{kj}^{-1} \tag{3A}$$

$$C_k(b,j) = \frac{C^*_k(b,j) + w}{1 + 4w} \tag{3B}$$

Here, $q_b$ denotes the frequency of base $b$ in the input sequence set, and $Z_{kj}$ is a column specific normalization constant chosen such that the column $j$ of base probability matrix $C_k$ sums to one. The first equation defines the MAP (maximum a posteriori probability) estimation of the base probability matrix for each class $k$. The second equation adds a small correction term to the MAP estimations that prevents probabilities from converging to zero. Note however, that the algorithms returns $C^*_j$ as the final results after the last iteration. A small correction term $x$ is also added to the re-estimated class probabilities:

$$p_k = \frac{\left(\frac{1}{N}\right)\left(\sum_i P(C_k|S_i)\right) + x}{1 + Kx} \tag{4}$$

The algorithm is initiated by a random seeding strategy. The probabilities of individual sequences of belonging to specific classes are sampled from a Beta distribution

$$P(C_k|S_i) \sim \frac{\text{Beta}(\alpha, \beta)}{Z_i} \tag{5}$$

with shape parameters $\alpha$=0.01 and $\beta$=1. $Z_i$ is a sequence-specific normalization constant chosen such that the class probabilities for sequence $i$ sum to one. The classes themselves are assigned equal probabilities $p_k$=1/$K$. After initializing these probabilities, the EM algorithm starts with Eq 3.

## Probabilistic partitioning extended algorithm

The extended partitioning algorithm (Fig 1B) features two innovations: (i) a two-state clustering strategy and (ii) a new, so-called "over-skewing" parameter $\sigma$. The two extensions are independent of each other, *i.e.* two-stage clustering can be used without over-skewing, and vice-versa. Two-stage clustering serves to increase the reproducibility of the results when initiating the algorithm with different random seeds. Over-skewing causes the algorithm to prefer partitionings with classes of highly unequal sizes, typically a majority class plus a number of small classes with highly skewed base compositions.

With the two-stage clustering strategy, the basic EM algorithm is applied $n$ times to produce $n \times K$ subclasses. Each subclass is characterized as a base probability matrix henceforth referred to as a "motif". During the second stage, the motifs from the first stage are hierarchically clustered and subsequently partitioned into motif groups using a fixed height $h$. The $K$ largest

motif groups are retained, and a consensus base probability matrix $C_k$ is computed for each group by averaging over all its members. Likewise, the $p_k$ is computed as the average over the occurrence probabilities of all motifs belonging to group $k$. Hierarchical clustering was carried out with the R functions *dist* and *hclust*, using "Euclidean" as a distance measure, and "complete" as a clustering method. Tree partitioning was carried out with the R function *cutree*.

## Computation of log-odds and Z-scores

The probability matrix for the human promoter class 3 (Fig 3, numerical data in S2 File) was used to compute "log-odds" and "Z-scores" for wild-type and mutant promoter sequences (Tables 1 and S3). Log-odds scores were computed by converting the probability matrix obtained from probabilistic partitioning into a weight matrix assuming a uniform base composition

$$w_{bi} = \ln\left(\frac{p_{bi}}{0.25}\right) \tag{6}$$

and then summing up the position-specific weights corresponding to a particular sequence. Z-scores were estimated by normalizing the log-odds scores with respect to the mean and standard deviation of the scores obtained from 100 shuffled versions of the same promoter sequence. Note that according to Berg and von Hippel [61], a log-odds score computed from observed base frequencies should be inversely proportional to the binding energy of the corresponding protein-DNA complex. Z-scores, on the other hand, serve to estimate the probability that a motif match found in a real sequence could be due to chance.

## ATAC-seq analysis

Average ATAC-seq footprints for promoter classes were produced with public data from human lymphoblastoid cell line GM12878 [62], and from *Drosophila* wild type eye-antennal imaginal disc [63], see supplementary material for GEO accession numbers and download URLs. We used processed versions of the data, *i.e.* read alignment files, available from the MGA repository [64]. These files were generated by mapping the raw reads to the dm6 and hg19 assemblies using bowtie2. The genomic positions corresponding to the 5' ends of the mapped reads were considered nucleolytic cleavage sites and thus used as reference points for computational footprint visualization. Cleavage sites on the + and - strand of the genome were shifted 4 bp downstream and upstream, respectively, for optimal superposition. The aggregation plots for the promoter classes shown in Fig 4 were generated via the web interface of the ChIP-Cor tool [65] using the following parameters: Reference feature *oriented*, target feature *any*, *centering* 4, window width 1, count cut-off 10, normalization *global*.

## CAGE and Chromatin architecture around promoters

*Homo sapiens* CAGE data was obtained from the FANTOM consortium [66] combining all samples together, whereas *Drosophila melanogaster* data was obtained from Machibase [67], combining all embryo samples. Given the presence of few highly expressed promoters that would skew the result interpretation, a cap of 1000 tags mapping to the same location was used. Correlation analysis was performed using ChIP-seq [65] with data already present in the MGA database [64].

 *Homo sapiens* nucleosome data was obtained from Gaffney et al. [68] whereas *Drosophila melanogaster* data was obtained from Chereji et al. [69] using the samples of 12h embryos with high MNase treatment. Correlation analysis was performed using ChIP-seq with data from the MGA database.

Nucleosome favoring signals were studied using OProf [70] scanning the region around promoters with a window of 12bp with the consensus sequence SSSNWWWNSSS allowing 2 mismatches. Fourier transform was performed in R using the function "spec.pgram".

## YY1 motif enrichment profiles of different human promoter classes

The motif enrichment profiles were generated with the TF binding site matrix MA0095.2 YY1 (Length=12) from JASPAR 2020 [71] with a window size of 16 and in unidirectional search mode. Numerical data (provided in S2 File) were generated using the web interface of the OProf program from the Signal Search Analysis server [70].

## Neighbor joining analysis

Promoter sets of 10 organisms (*H. sapiens*, *M. musculus*, *D. rerio*, *C. elegans*, *D. melanogaster*, *A. mellifera*, *A. thaliana*, *Z. mays*, *S. cerevisiae*, *S. pombe*) were analyzed with the newly developed algorithm. In the first step (Fig 1A), 200 iterations were applied by the probabilistic partitioning to generate 6 motifs. This procedure was independently repeated 50 times to generate 300 motifs for each species (see Fig 1A for reference). The motifs were then hierarchically clustered, and the resulting tree was cut to obtain 10 clusters (Fig 1B). The 6 nodes with the highest number of motifs were then chosen and averaged to generate the final motifs. These motif collections were further clustered with Euclidean distance (function 'dist', from package 'stats') and plotted using a Neighbor Joining tree (function 'nj' from package 'ape' [72]). The frequency matrices of motifs belonging to each of the 3 branches were averaged to generate the branch consensus.

## Plasmid construction

For cloning the minimal promoters of the selected genes into a reporter plasmid, double-stranded oligonucleotides (IDT) comprising core promoter sequences from −10 to +40/+41 were inserted into the KpnI and SpeI sites of a pGL3-Basic plasmid with a modified polylinker. For each promoter, both WT and mutated preferred downstream positions (mPDP) (G>T at positions +24, +28 and +29 relative to the relevant $A_{+1}$ position) versions were cloned. Primers used are listed in S1 File. All generated constructs were verified by sequencing (Hy Labs).

## Cell culture, transient transfections and reporter gene assay

Human Embryonic Kidney (HEK) 293 cells were cultured in DMEM high-glucose (Biological Industries) supplemented with 10% FBS, 0.1% penicillin-streptomycin, and 1% L-Glutamine, and grown at 37˚C with 5% $CO_2$.

For dual luciferase assays, 1-2x$10^6$ cells were plated per 60mm dish one day prior to transfection. Cells were transfected using the calcium phosphate method with a total of 3μg DNA (2.5μg firefly luciferase plasmid, 100ng of Thymidine Kinase-*Renilla* luciferase plasmid, and 400ng of pBlueScript plasmid) per 60mm dish. Prior to the transfection, the medium was changed to contain 25μM Chloroquine, and replaced with fresh medium 6-8 hours following the transfection. Cells were harvested 48 hours post-transfection and assayed for dual-Luciferase activities as specified by the manufacturer (Promega). To correct for variations in transfection efficiency, the firefly luciferase activity of each sample was normalized to the corresponding *Renilla* luciferase activity. Each transfection was performed in triplicates, and each graph represents an average of 4 to 6 independent experiments ± SEM. Student's two-sided t-test was applied in order to determine the statistical significance of the observed difference.

## Supporting information

**S1 File. Primers used to generate the examined promoters.**
(XLSX)

**S2 File. Numerical values for all relevant figures.**
(XLSX)

**S1 Table. Three bp periodic distributions of ATG in human promoter classes 4-6.**
(DOCX)

**S2 Table. Distribution of promoter classes with TATA-box status and "shape" status (narrow/broad).**
(DOCX)

**S3 Table. The EM algorithm (this study) makes similar predictions as the SVRb model.**
(DOCX)

**S1 Fig. DPE distribution around *Drosophila melanogaster* and *Homo sapiens* promoters.**
DPE motif distribution around all *D. melanogaster* and *Homo sapiens* promoters was calculated using the DPE logo depicted in the top left corner. The "shuffle" distribution is derived by scanning shuffled sequences with the DPE motif and should be considered as a background signal. Numerical data (provided in S2 File) were generated using the web interface of the OProf program from the Signal Search Analysis server [70].
(PDF)

**S2 Fig. Partitioning of *Drosophila* promoter sequences with the extended EM algorithm.**
Numerical data are provided in S2 File.
(TIF)

**S3 Fig. Bootstrap analysis of human promoter classes.** The complete promoter sequence collection was resampled 10 times. The extended partitioning algorithm was applied to the bootstrapped data sets retaining the 10 most frequently found classes. The heatmap reflects the similarity (expressed as Pearson correlation coefficients) of the newly identified motifs with the corresponding most similar motifs found in each bootstrapping round.
(PDF)

**S4 Fig. The promoter classes identified by EM algorithms are enriched for distinct GO terms.** Gene lists comprising the *Drosophila melanogaster* classes identified by the basic probabilistic partitioning EM algorithm (Fig 2) and the human classes identified by the extended probabilistic partitioning EM algorithm (Fig 3), was analyzed using PANTHER-GO slim Biological Process annotation data set [45]. The enrichment scores are presented as $-\log_{10}(P$ value). GO terms enrichment is indicated by blue bars, whereas GO terms depletion is indicated by red bars. The Inr-containing *Drosophila* Class 2 and the weak non-canonical Inr motif-containing Class 3 are enriched for metabolic and biological processes. Class 1 of human promoters identified using the new extended algorithm is depleted for immune response. As promoters in this class account for 87.9% of promoters of the human promoters, the significance of this depletion is unclear. Numerical data are provided in S2 File.
(TIF)

**S5 Fig. YY1 motif enrichment profiles of different human promoter classes.** The motif enrichment profiles were generated with the TF binding site matrix MA0095.2 YY1 (Length=12) from JASPAR 2020 [71] with a window size of 16 and in unidirectional search mode. Due to the small window size, the height of the peaks does not reflect the total fraction

of promoters that contain a YY1 motif in the corresponding promoter class. Numerical data (provided in S2 File) were generated using the web interface of the OProf program from the Signal Search Analysis server [70].
(PDF)

**S6 Fig. EPDnew screenshots of the analyzed promoters, used to define promoter shape.** FANTOM5-generated CAGE tags distribution of individual promoters in HEK-293 cells was manually examined using the EPDnew viewer, in order to determine their transcription initiation pattern.
(PDF)

**S7 Fig. Comparison of experimental logos with sequence motif logos.** (A) Experimental logos are based on exhaustive single-base mutational analysis of the +15 to +29 region of two *Drosophila* promoters [22]. Relative expression values were rescaled such as to sum up to one at each position. The sequence motif logos were extracted from the logos shown in Figs 2 and 3. All logos have been over-skewed with an exponent of 2 to highlight differences between them. (B) Correlation plot showing Pearson correlation coefficients computed from the base probabilities underlying the logos. Note the high correlation of the *CG10479* experimental logo with the *Drosophila* motif logo. Numerical data are provided in S2 File.
(PDF)

# Acknowledgments

We are indebted to Jim Kadonaga for his generous support and invaluable suggestions, and for sharing data prior to publication. We thank Jim Kadonaga, Yehuda M. Danino, Orit Adato, Hadar Krap and Hodaya Komemi for critical reading of the manuscript.

# Author Contributions

**Conceptualization:** René Dreos, Anna Sloutskin, Philipp Bucher, Tamar Juven-Gershon.

**Data curation:** René Dreos, Anna Sloutskin.

**Formal analysis:** René Dreos, Anna Sloutskin, Nati Malachi, Diana Ideses, Philipp Bucher, Tamar Juven-Gershon.

**Funding acquisition:** Anna Sloutskin, Philipp Bucher, Tamar Juven-Gershon.

**Investigation:** René Dreos, Anna Sloutskin, Nati Malachi, Diana Ideses, Philipp Bucher, Tamar Juven-Gershon.

**Methodology:** René Dreos, Anna Sloutskin, Philipp Bucher.

**Project administration:** Philipp Bucher, Tamar Juven-Gershon.

**Resources:** René Dreos, Philipp Bucher.

**Software:** René Dreos, Philipp Bucher.

**Supervision:** Philipp Bucher, Tamar Juven-Gershon.

**Validation:** René Dreos, Anna Sloutskin, Nati Malachi, Diana Ideses, Philipp Bucher.

**Visualization:** René Dreos, Anna Sloutskin, Philipp Bucher, Tamar Juven-Gershon.

**Writing – original draft:** René Dreos, Anna Sloutskin, Philipp Bucher, Tamar Juven-Gershon.

**Writing – review & editing:** René Dreos, Anna Sloutskin, Nati Malachi, Diana Ideses, Philipp Bucher, Tamar Juven-Gershon.

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
