## [Decision Letter · Decision Letter 0]

2 Mar 2021

Dear Prof. Juven-Gershon,

Thank you very much for submitting your manuscript "Computational identification and experimental characterization of preferred downstream positions in human core promoters" for consideration at PLOS Computational Biology.

As with all papers reviewed by the journal, your manuscript was reviewed by members of the editorial board and by several independent reviewers. In light of the reviews (below this email), we would like to invite the resubmission of a significantly-revised version that takes into account the reviewers' comments.

We cannot make any decision about publication until we have seen the revised manuscript and your response to the reviewers' comments. Your revised manuscript is also likely to be sent to reviewers for further evaluation.

Sincerely,

Denis Thieffry, PhD

Associate Editor

PLOS Computational Biology

William Noble

Deputy Editor

PLOS Computational Biology

Reviewer's Responses to Questions

**Comments to the Authors:**

**Reviewer #1: **

Summary:

The authors investigated the structure of human core promoters with respect to the well-characterized Drosophila downstream promoter element (DPE), which is associated with an initiator element (Inr) located at a defined distance upstream from it. Based on some previous evidence, they hypothesized that DPE could exist in a fraction of human promoters and developed an original method of pattern discovery based on probabilistic partitioning algorithms by maximizing expectations which they adapted to extract patterns present in small sets of promoters. Among the discovered patterns, one looks similar to the Drosophila element Inr+DPE which they also exclusively found in other metazoan species. Using dual-luciferase assays, the authors showed a clear impact of DPE site mutations on promoter activity and a less obvious effect of spacer alteration. Overall, this study, which is nicely reported and pleasant to read, accessible and interesting for both bioinformatician and biologist communities, proposes a new well described methodology that could be generally applied when searching for complex motifs present in small but biologically relevant subsets of regions which are hardly detectable by classic methods. The functionality of the newly predicted human DPE is well supported by mutation assays, however, the need for strict spacing between Inr and DPE elements is less robust and would require further investigation.

Major issues:

1/ The authors selected 5 genes upon two scores, one from their motif, the second from a motif derived from a published experimental work on Drosophila. Those scores substantially vary between the genes, for instance one of them has a negative score using their motif, which, to my understanding, would not represent a potential human Inr+DPE candidate. The authors should clarify the criteria applied for this selection.

2/ The authors evaluated the importance of the spacing between the Inr and the putative DPE in 2 of these genes by adding or deleting 2 nucleotides at 2 different positions, and reported a consistent decrease of activity for one of the genes, while the readout is less clear for the other which show higher variability between replicates and non significant decrease for the deletion at position 10 which makes the promoter activity of this gene not strictly relying on the spacing. The analysis of a third candidate may reinforce their conclusion.

3/ In the discussion, three new results appear that were not present neither in the Results section nor in the methods.

- the first one concerns the functional enrichment of genes belonging to the motif classes 2 and 3 (l. 492, l. 553)

- the second reports a comparison of discovered Inr+DPE motifs in human and Drosophila with experimentally derived MTE motifs from 2 Drosophila genes (l. 509-531)

- the third corresponds to the ATG phasing with the 3 motif classes showing CGN repeats (l. 570).

4/ The authors applied the basic EM algorithm on eight eukaryotes species, and found an exclusive presence of Inr+DPE motifs only in metazoan. However, the use of the extended algorithm could have highlighted minor motifs in non-metazoan species as it allowed to detect Inr+DPE motifs in human.

Minor issue:

The ATAC-seq profiles are not easy to interpret and would gain in clarity when coming to comparison by joining the profiles in one plot either by species or all together (using line plots for instance) enlarging by this way the x-axis which would help to spot the precise positions which are the main points of the graphs.

**Reviewer #2: **

The manuscript by Dreos et al describe a computational discovery and experimental validation of long-elusive mammalian equivalent of the Drosophila downstream promoter element (DPE) in human promoters.

The known features of Drosophila DPE and the promoters that contain it are:

- It is present at nearly 30% of the promoters of protein-coding genes.

- It is highly overrepresented in genes associated with development and developmental gene regulation.

- It is associated with focused promoters, with a fixed (to a single bp) spacing relative to their preferred transcription start site.

On the other hand, the human DPE that the authors report has the following features:

- It is present at around 3% of promoters of protein-coding genes. While the promoters that contain it hav a better-than average defined Inr, the DPE motif itself is weak enough (cca 2 bits of information) that it can probably be found at more than 3% of arbitrary positions within the promoter.

- The genes with DPE-containing promoters show no association with development or gene regulation: the authors mention this in passing in Discussion, without showing analysis to support this (admittedly negative) observation.

- It is unclear whether this set of promoters is mostly focused (no such analysis is presented), although the authors hand-pick focused candidates for experimental validation.

Given the significant differences, the authors dwell surprisingly little on the causes for them. In Drosophila , Inr+DPE promoters are a well defined functional and architectural class associated with genes responsive to multiple enhancers, including those at long distances from the promoter (as shown by Stark lab STARR-seq papers). This has either been acquired since the last common ancestor of insects and vertebrates, or it was an ancestral state that has subsequently been largely supplanted by other promoter type in vertebrates, leaving only a small minority of promoters with DPE-dependent architecture. Either that, or the mammalian DPE, while superficially similar to the Drosophila one, is something different altogether. (The results in the recent paper from Kadonaga lab, while stronger on experimental validation, are also amenable to the same questions.)

Specific comments:

- Drosophila DPE promoters, like TATA promoters in all Metazoa so far, are motif-dependent and, unlike dispersed promoters, do not feature a stably positioned downstream nucleosome. How does the human DPE promoter class look in that regard? Does it a) have a strong nucleosome position, which would argue for nucleosome-dependent transcriptional initiation, another major difference from the Drosophila DPE, b) no strong nucleosome position, which would lead one to expect that it is always associated with focused transcription start site, or 3) does the nucleosome position depend, even within this class, on whether transcription initiation is focused or dispersed?

- In case of stable nucleosome position, could the validation experiments that remove the most informative DPE motif positions or change the spacing between DPE and TSS be interpreted via their effect on the nucleosome positioning? For example, if a nucleosome is shifted 2bp left or right, the optimal position of the Inr would shift with it, and if there is no optimal Inr motif 2bp left or right, the efficiency of initiation would decrease? The problem with such functional probing is that there is no guarantee that the changes authors introduce have anything to do with the DPE motif itself, or that they wouldn't give similar results on other promoters with similar TSS dispersion and nucleosome positioning properties. That also holds for the three human promoters apparently historically demonstrated to have functional DPE (IRF1, CALM2, TAF7).

- The decreases in promoter activity upon mutating away the proposed DPE motif are relatively mild. If it is an essential part of the architecture of those promoters like TATA (as opposed to e.g. providing some gentle cues), shouldn't the effect be much more dramatic? Doesn't the abolishment of DPE in Drosophila developmental promoters have a much more pronounced effect?

- Partitioning of Drosophila and human promoters using basic (Fig 2) and human promoters using new extended EM algorithm (Fig 3): The obtained classes should be checked for association with other promoter elements outside the (-10,+40) region, to see how they correspond to previously proposed promoter types (which ones have TATA, which ones have nucleosome positioning signal, etc.). Also, the position of proximal downstream YY1 motif should be of interest - it is present at about the same position, has some similarity to the discovered DPE motif, and clips the extent of the dispersed TSS. An older paper (https://genome.cshlp.org/content/17/6/798 ) has found both YY1 and the novel "motif8" element at approximately the same position.

- The ATAC-seq footprints in Fig 4 look strange - I do not understand how "single-base resolution" footprints are obtained - was it just by taking the midpoints of the fragments? In such case a "V-plot" would be more interesting, as it would reveal the positions of nucleotide-size fragments and help answer at least some of the above questions about nucleosome positioning in different classes.

- Regarding the super-core promoter: is there any evidence that TATA box and DPE in it are used simultaneously instead of being mutually exclusive (i.e. either one or the other is used in a particular transcription initiation event but not both)?

**Reviewer #3: **

In this paper Dreos, et al, develop a method to identify patterns in core promoters using a modified expectation maximization algorithm (EM). The method enables the identification of hidden patterns that have been profoundly characterized in Drosophila but that have scaped clear determination in human promoters.

Core promoters are defined as the -40 +40 regions around transcriptional start sites, these regions are bound by the RNA polymerase machinery and while clear patterns of sequences required for this function have been characterized in Drosophila, their characterization in other organisms has been challenging.

Authors apply a probabilistic partition algorithm based in expectation maximization to enable discovery of highly variable but conserved sequences in core promoters in human and Drosophila. Authors recapitulate previously known biologically meaningful patterns in the Drosophila data set and are able to discover previously known and new biology in human promoters. Particularly interesting, the new insight in patterns in human promoters are similar to patterns in Drosophila. Moreover, the results of the program in other organisms reveals a conservation pattern across species.

Promoter predictions are then experimentally tested and the bases that were predicted to be more relevant in the promoter function are the ones that when mutated have a higher impact on the promoter function.

2) Specific comments for revision:

a) Major.

The EM algorithm is shown to be able to recover relevant motifs in the promoters, however, for the selection of promoters to be tested experimentally authors use the tool ElemeNT with the motifs provided by that tool, instead of using the motifs discovered by EM. For this reviewer is not clear why this decision was made, I understand the tools ElemeNT has been implemented to run pattern-matching looking for positional preferences, but I would expect the usage of the motifs discovered and described in this article and not others. Is important that this is clarified in the text.

b) Minor

Figure 5A: Maintain negative positions as reference to the previous figure4.

Line 258 should be revised

**Have all data underlying the figures and results presented in the manuscript been provided?**

Reviewer #1: Yes

Reviewer #2: Yes

Reviewer #3: Yes

PLOS authors have the option to publish the peer review history of their article (what does this mean?). If published, this will include your full peer review and any attached files.

Reviewer #1: **Yes: **Elodie Darbo

Reviewer #2: No

Reviewer #3: No
---

## [Decision Letter · Decision Letter 1]

15 Jun 2021

Dear Prof. Juven-Gershon,

Thank you very much for submitting your manuscript "Computational identification and experimental characterization of preferred downstream positions in human core promoters" for consideration at PLOS Computational Biology. As with all papers reviewed by the journal, your manuscript was reviewed by members of the editorial board and by several independent reviewers. The reviewers appreciated the attention to an important topic. Based on the reviews, we are likely to accept this manuscript for publication, providing that you modify the manuscript according to the review recommendations.

As the reviewers ask only for minor adjustments and clarifications, we ask you to prepare and submit your revised manuscript within 30 days. If you anticipate any delay, please let us know the expected resubmission date by replying to this email.

Sincerely,

Denis Thieffry, PhD

Associate Editor

PLOS Computational Biology

William Noble

Deputy Editor

PLOS Computational Biology

[LINK]

Reviewer's Responses to Questions

**Comments to the Authors:**

Reviewer #1: 

The authors seriously considered our previous remarks and largely assessed the highlighted issues. This significantly improved the paper. However, the score definitions (lod-odds and z-scores) and its use remain somehow unclear to me.

The authors explain that the log scores are strongly influenced by base composition, so can we rely on this score? Is the z-score a better estimation than the log-odd then? If a user was about to use the EM algorithm on its favorite species and want to test some of its hits, what the authors would recommend? I may have missed the information but a clear definition of the scores, their potential bias and how to interpret them could help future users.

Reviewer #2: 

The revised version of the manuscript by Dreos et al. has addressed many of my original concerns, and added a number of new analyses that resulted in interesting observations.

While the evidence still does not prove conclusively that the human Inr+DPE promoter shares the same evolutionary origin as the Drosophila one (3% vs 30% genes that contain it, and no clear functional preference vs clear developmental specialisation are very important differences still), it provides a lot of observations for future study. I am especially excited about the differences in YY1 binding across classes.

Reviewer #3: 

I consider the authors have addressed all concerns and substantially improved their articule.

I just noticed that they forgot to add the methods for the YY1 analysis that was included during revisions, please include them so the manuscript can be completed.

**Have the authors made all data and (if applicable) computational code underlying the findings in their manuscript fully available?**

Reviewer #1: Yes

Reviewer #2: Yes

Reviewer #3: Yes

PLOS authors have the option to publish the peer review history of their article (what does this mean?). If published, this will include your full peer review and any attached files.

Reviewer #1: No

Reviewer #2: No

Reviewer #3: No

Figure Files:

Data Requirements:

Reproducibility:

References:

---

## [Editor Report · Decision Letter 2]

7 Jul 2021

Dear Prof. Juven-Gershon,

We are pleased to inform you that your manuscript 'Computational identification and experimental characterization of preferred downstream positions in human core promoters' has been provisionally accepted for publication in PLOS Computational Biology.

Best regards,

Denis Thieffry, PhD

Associate Editor

PLOS Computational Biology

William Noble

Deputy Editor

PLOS Computational Biology

---

## [Editor Report · Acceptance letter]

6 Aug 2021

PCOMPBIOL-D-20-01849R2 

Computational identification and experimental characterization of preferred downstream positions in human core promoters

Dear Dr Juven-Gershon,

I am pleased to inform you that your manuscript has been formally accepted for publication in PLOS Computational Biology. Your manuscript is now with our production department and you will be notified of the publication date in due course.

With kind regards,

Agota Szep
